# Ornamental Plant Efficiency for Heavy Metals Phytoextraction from Contaminated Soils Amended with Organic Materials

**DOI:** 10.3390/molecules26113360

**Published:** 2021-06-02

**Authors:** Mahrous Awad, M. A. El-Desoky, A. Ghallab, Jan Kubes, S. E. Abdel-Mawly, Subhan Danish, Disna Ratnasekera, Mohammad Sohidul Islam, Milan Skalicky, Marian Brestic, Alaa Baazeem, Saqer S. Alotaibi, Talha Javed, Rubab Shabbir, Shah Fahad, Muhammad Habib ur Rahman, Ayman EL Sabagh

**Affiliations:** 1Department of Soils and Water, Faculty of Agriculture, Al-Azhar University, Assiut 71524, Egypt; saberassiut46@gmail.com; 2Department of Soils and Water, Faculty of Agriculture, Assiut University, Assiut 71524, Egypt; mohamed.zedan@aun.edu.eg; 3Department of Soils and Water, Faculty of Agriculture, Aswan University, Aswan 81711, Egypt; ahmed.ibrahim3@agr.au.edu.eg; 4Department of Botany and Plant Physiology, Faculty of Agrobiology, Food and Natural Resources, Czech University of Life Sciences Prague, Kamycka 129, 16500 Prague, Czech Republic; kubes@af.czu.cz (J.K.); skalicky@af.czu.cz (M.S.); marian.brestic@uniag.sk (M.B.); 5Department of Soil Science, Bahauddin Zakariya University, Multan 06110, Pakistan; sd96850@gmail.com; 6Department of Agricultural Biology, Faculty of Agriculture, University of Ruhuna, Peradeniya 20400, Sri Lanka; disnaratnasekera@gmail.com; 7Department of Agronomy, Hajee Mohammad Danesh Sience and Technology University, Dinajpur 5200, Bangladesh; shahid_sohana@yahoo.com; 8Department of Plant Physiology, Slovak University of Agriculture, Nitra, Tr. A. Hlinku 2, 94901 Nitra, Slovakia; 9Department of Biology, College of Science, Taif University, P.O. Box 11099, Taif 21944, Saudi Arabia; aabaazeem@tu.edu.sa; 10Department of Biotechnology, College of Science, Taif University, P.O. Box 11099, Taif 21944, Saudi Arabia; saqer@tu.edu.sa; 11College of Agriculture, Fujian Agriculture and Forestry University, Fuzhou 350002, China; mtahaj@fafu.edu.cn (T.J.); rubabshabbir28@gmail.com (R.S.); 12Agriculture Department, The University of Swabi, Khyber Paktunkhwa 94640, Pakistan; shahfahad@uoswabi.edu.pk; 13Crop Science Group, Institute of Crop Science and Resource Conservation (INRES), University Bonn, 53115 Bonn, Germany; mhabibur@uni-bonn.de; 14Department of Agronomy, MNS-University of Agriculture, Multan 60000, Pakistan; 15Department of Agronomy, Faculty of Agriculture, Kafrelsheikh University, Kafr El-Shaikh 33516, Egypt

**Keywords:** contaminated soil, heavy metals, ornamental plants, phytoremediation, organic materials

## Abstract

Accumulation of heavy metals (HMs) by ornamental plants (OPs) from contaminated agriculture soils is a unique technique that can efficiently reduce the metal load in the food chain. *Amaranthus tricolor* L. has attractive characteristics acquiring a higher growth rate and large biomass when grown at heavy metal contaminated soils. Site-specific detailed information is not available on the use of *A. tricolor* plant in metal phytoremediation from the polluted sites. The study aimed to enhance the uptake of HMs (Pb, Zn, and Cu) via amending poultry litter extract (PLE), vinasse sugarcane (VSC), and humic acid (HA) as natural mobilized organic materials compared to ethylene diamine tetraacetic acid (EDTA), as a common mobilized chemical agent by *A. tricolor* plant. The studied soils collected from Helwan, El-Gabal El-Asfar (Cairo Governorate), Arab El-Madabeg (Assiut Governorate), Egypt, and study have been conducted under pot condition. Our results revealed all organic materials in all studied soils, except EDTA in EL-Gabal El-Asfar soil, significantly increased the dry weight of the *A. tricolor* plant compared to the control treatment. The uptake of Pb and Zn significantly (*p* > 0.05) increased due to applying all organic materials to the studied soils. HA application caused the highest uptake as shown in Pb concentration by more than 5 times in Helwan soil and EDTA by 65% in El-Gabal El-Asfar soil while VSC increased it by 110% in El-Madabeg soil. Also, an increase in Zn concentration due to EDTA application was 58, 42, and 56% for Helwan, El-Gabal El-Asfar, and El-Madabeg soil, respectively. In all studied soils, the application of organic materials increased the remediation factor (RF) than the control. El-Madabeg soil treated with vinasse sugarcane gave the highest RF values; 6.40, 3.26, and 4.02% for Pb, Zn, and Cu, respectively, than the control. Thus, we identified *A. tricolor* as a successful ornamental candidate that, along with organic mobilization amendments, most efficiently develop soil health, reduce metal toxicity, and recommend remediation of heavy metal-contaminated soils. Additionally, long-term application of organic mobilization amendments and continued growth of *A*. *tricolor* under field conditions could be recommended for future directions to confirm the results.

## 1. Introduction

Contaminated soils by heavy metals (HMs) are the global environmental problem caused by certain natural processors, including erosion, mineral weathering, volcano eruptions and anthropogenic activities such as industrialization, urbanization, agricultural practices (application of HMs-containing pesticides and fertilizers, sewage sludge, etc.) and casting minerals [1,2,3,4,5,6]. Their accumulation in agricultural soils could potentially increase the risk of entry into the food/feed chain, threatening human and animal health [7,8]. For cleaning-up of HM contamination, various practices and technologies such as adsorption, oxidation, reduction, precipitation [9], ion-exchange, coagulation-flocculation, electrochemical methods [10], leaching/acid extraction, and soil washing [11] have been adopted. However, such remedial methods are not economically viable, destructive to the agro-ecosystems, impractical for large quantities of hazardous waste [12]. Also, changing the metal mobility via immobilization or mobilization with soil amendments and green remediation has been recognized as a calm remediation tactic [13].

The use of plants for remediation of HM contaminations attracts special attention because of additional beneficial effects, including preventing erosion, improving soil quality, and maintaining healthy ecosystem functioning. Further, it is an economical and easily applicable eco-friendly and natural process, wherein no residual and toxic materials are left behind, preserve, maintain soil physical, chemical, and biological properties, and prevents HMs penetration into groundwater [14]. The phytoremediation should be a holistic approach to immobilize or lower the HM from the contaminated soil and restore the healthy soil characteristics to perform its normal functions. The success of this practice depends on the ability of the selected plant to produce rapid biomass and accumulate HMs within their tissues [15,16]. Plant mediating to absorb, transfer and accumulate hazardous contaminants using edible plants [17], non-edible plants including medicinal and aromatic species [18,19] along with trees, grasses, and ornamental plants [20] has been documented. Using edible crops is an inappropriate option because pollutants enter the food chain, threatening human and animal health [21,22].

As a suitable option, ornamental plants (OPs) may be of special interest because of their fast growth, viability, abundance, thus preferably chosen from the local environment. The characteristics to be considered when selecting an OP were well documented [23]. Among these species, *Amaranthus tricolor*, also known as red amaranth, which belongs to the *Amaranthaceae* family, originated in Asia would be one of the best candidates. It is the most common cultivated plant in Bangladesh as leafy vegetables [24]. Also, Ref. [25] identify the *A. tricolor* as the Cd hyperaccumulators. Moreover, these plants can clean up the contaminants, beautifying the environment, and pollutants without access to the food chain [24,26]. It tends to accumulate HMs in their non-food biomasses that significantly contribute to their economic and ecological values [22,27].

The factors affecting the amounts of metal that a plant absorbs those controlled by the concentration and speciation of the metal in the soil solution; the movement of metal from the bulk soil to the root surface; the transport of the metal from the root surface into the root, and its translocation from the root to the shoot [28,29]. Plants readily uptake HMs dissolved in soil solution in ionic or chelated, or complex forms [30,31]. Several organic and inorganic agents might effectively and specifically increase HMs solubility that could be accumulated by several plant species [32]. Ethylenediaminetetraacetic acid (EDTA), as a synthetic chelator, is used among the most common chelating agents to increase the plant uptake of metals cause environmental contamination by leaching HMs into groundwater bodies [7]. Further, low biomass of hyperaccumulation increased bioavailability of metals, and the persistence of metal-chelate complexes are major disadvantages of such mobilizing synthetic chelators [33].

As alternatives to synthetic chelators, widespread natural sources called biochelators, such as poultry litter extract (PLE), vinasse, and humic substances, could be used. Enhanced soil biological, chemical, and physical properties have been reported with increased soil fertility by applying such organic amendments [34]. The organic materials, including poultry litter, Vinasse, humic acid are by-products or industry wastes with various unique benefits related to soil quality and health [31,34,35]. Using such organic materials and different plant species to boost phytoextraction was previously reported [16,20,21]. For example, Ref. [36] pointed out that biochelators (humic acids) realized positive effects on the HMs uptake. They also found that adding humic acids (HA) to artificially contaminated soil increased the cadmium (Cd) uptake by tobacco plants. Moreover, the application of soil amendments such as biosolids and cow manure exhibited the enhanced uptake HMs in sunflower [16]. Exploring the phytoextraction potential of ornamental plants for remediation of polluted environments is considered an important strategy that could be highlighted in future research [11,14].

Particularly *A**. tricolor* was reported to have specific interest amongst other ornamental plants as it has specific mechanisms to solubilize Cd adsorbed to soil particles while acquiring higher growth rate and large biomass when grown in heavy metal contaminated soils [25]. Hence, *A. tricolor* is a suitable model to explore metal transfer and phytoremediation of HMs grown in contaminated soils amended by different organic and inorganic materials. Most of the previous studies evaluate the performance of one or two organic or synthetic materials on contaminated soil, mostly using edible plants to assess their study. Our hypothesis that each material or metal may have specific behaviour depending on the soil characteristics or metal and the ability of the used biomaterial to make the metal more soluble. In our study, we considered the advantage of using three natural materials that are already present in a huge quantity, problematic in getting rid of them and imposing additional cost in waste management programs of the country. Therefore, the purpose of this study was to compare the performance of naturally occurring organic materials (poultry litter extract (PLE), vinasse sugarcane (VSC), and humic acid (HA)) with synthetic chelates (EDTA) in enhancing phytoextraction of lead (Pb), zinc (Zn) and copper (Cu) by *A. tricolor* plants in different naturally contaminated soils, and to assess the storing and removing of metals using ornamental plants and their ability to produce a large amount of biomass in a short time for such soil characteristics.

## 2. Materials and Methods

### 2.1. Characterization of Soil Sites

The surface soil layer (topsoil) (0–30 cm) was collected from three different locations in Egypt (Helwan and El-Gabal El-Asfar, Cairo Governorate, and El-Madabeg soil, Assiut governorate). The soils at selected sites are continuously contaminated by HMs from sewage (El-Gabal El-Asfar and El-Madabeg soils) and/or industrial (Helwan) wastewaters. The soils at these locations are receiving a continuous supply of heavy metals as domestic, such as the soils of El-Gabal El-Asfar (31°23′15″ E and 30°12′19″ N) and El-Madabeg (31°08′42″ E and 27°09′52″ N) for more than 50 years, and the Helwan soil (31°20′09″ E and 29°34′55″ N), which are very close to the army factories that receive their industrial waste in addition to human waste. The chosen soils varied in their texture between silty clay loam, loamy sand and sand, soil pH (8.11, 6.71 and 7.59) and organic matter (2.18, 5.70 and 2.80), respectively, for Helwan, El-Gabal El-Asfar and El-Madabeg soils (Figure 1).

The collected soil samples were air-dried, ground with a wooden roller, sieved to pass through a 2 mm sieve, and kept for experimental purposes. Available Pb, Zn and Cu were extracted from the soil samples using a 0.005 M of diethylene triamine pentaacetic acid (DTPA) solution buffered at pH 7.3 as described by [37]. A soil sample of 1 g was used to extract metal content using HNO_3_, H_2_O_2_, and HCl [38]. Certain physical properties, soil reaction (pH), soil salinity (EC), and soil organic matter of tested soils were measured according to [39] with three replicates for each and are presented in Table 1.

### 2.2. Testing Organic Materials

Four organic materials were examined for their effects on HMs mobility and phytoextraction capacity by growing tested plants on contaminated soils. Ethylenediaminetetraacetic acid (EDTA) at 2 mM as a synthetic organic material, in addition to poultry litter extract (PLE) solution (75 g/L), vinasse sugarcane (VSC) 1:2 water (*V*/*V*), and humic acid (HA) at 0.025% solution as natural organic materials was assembled as treatments. The investigated levels of the organic materials were chosen as effective levels according to the leaching procedures. The poultry litter (PL) was collected from the poultry farm of Assiut University, Assiut Governorate. Vinasse sugarcane (VSC), a by-product of the sugar industry, was obtained from Abu-korkas Sugar Factory, El-Minya Governorate, Egypt. The humic acid (HA) solution was brought from the Agriculture Company for Recycling Agriculture Residues, El-Minya Governorate, Egypt. PLE, VSC, and HA samples were subjected to necessary chemical analysis (OM, pH, and EC). Other samples were digested using concentrated nitric and perchloric acids to determine their total Pb, Zn, and Cu contents using atomic absorption (Table 2). The analysis of these organic materials was done in three replicates for each (*n* = 3).

### 2.3. Experimentation

A pot experiment under a greenhouse condition was conducted to study the ability of *A. tricolor* to accumulate Pb and Zn from contaminated soil. Plastic pots were filled with 2 kg of soil and two seedlings of *A. tricolor* (15-day-old) were grown in each pot. The pots were carefully watered to near field capacity by deionized water for two weeks. Four organic amendments were used in the current study (EDTA, PLE, VSC, and HA solutions). The pots were set up in a completely randomized design (CRD) and each treatment was replicated three times. At the end of the experiment (10 weeks from transplanting), plant samples were collected and cleaned first using tap water, followed by washing with distilled water. Then samples were oven-dried at 70 °C for 72 h, and the dry weight was recorded. The dried plant materials were ground using a mortar and pestle and kept for plant analysis with three replicates for each. Soil samples from each pot were air-dried, ground, and passed through a 2 mm sieve. Plant samples of 0.2 g were digested using concentrated acids of H_2_SO_4_ and HClO_4_. The heavy metals: Pb, Zn, and Cu in the digests were determined using model 906 of GBC atomic absorption spectrophotometry (AA 800, Perkin Elmer Co., 14775 E Hinsdale Ave, Centennial, CO, USA) according to the method outlined by [38] method (3050).

In brief, one gram of a soil sample is transferred to a digestion vessel, 10 mL of 1:1 HNO_3_ is added, mixed, and covered with a watch glass, heated to 95 ± 5 °C and refluxed for 10 to 15 min without boiling and allowed to cool. 5 mL of concentrated HNO_3_ is repeated and refluxed for 30 min until no brown fumes are given. After the last step is completed and the sample is cooled, add 2 mL of water and 3 mL of 30% H_2_O_2_. One mL of 30% H_2_O_2_ is added while warming until the minimal effervescence or until the total of 10 mL 30% H_2_O_2_ (should not be exceeded). Then, reduce the volume to approximately 5 mL. After that, 10 mL of concentrated HCl is added to the sample digest, heated for 15 min. During this whole process, the sample should be covered with a watch glass. The digestate is filtered through a Whatman No. 41 filter paper; the filtrate is collected in a 100-mL volumetric flask and made to the volume. This method successively digests a soil sample with HNO_3_, H_2_O_2_ and HCl acids to measure “total sorbed metals” by the acidic dissolution of clay, oxides, and carbonates and oxidation of organic matter; elements associated with silicates are not dissolved. All soil and plant sample measurements were performed in three replications.

### 2.4. Calculation of Remediation Factor

The percentage of remediation factor (RF) was calculated according to the equation [40]:(1)$$\mathrm{Remediation}\;\mathrm{Factor}\;\left(\mathrm{RF}\right)=\frac{\mathrm{Metal}\left(\mathrm{shoot}\right)\times \;\mathrm{DW}\times 100}{\mathrm{Metal}\left(\mathrm{soil}\right)\times \;\mathrm{weight}\;\mathrm{of}\;\mathrm{soil}\;\mathrm{in}\;\mathrm{the}\;\mathrm{pot}}$$
where, metal (shoot) is the concentration of metal in the shoot as milligram per gram, DW refers to the dry weight of shoot as gram per pot, and metal (soil) is the concentration of total soil Metal as milligram per kilogram.

### 2.5. Analysis of Data

One-way analysis of variance (ANOVA) and Duncan’s multiple range test was used to determine the statistical significance of the organic materials treatment effects on heavy metals uptake and plant development using CoStat software, and *p* < 0.05 was considered statistically significant. All the results are shown as mean values (*n* = 3) ± Standard deviation (SD).

## 3. Results

### 3.1. Effect of Organic Materials on Fresh and Dry Weight

The illustrated results in Figure 2a,b showed the application effect of the investigated organic materials on the fresh and dry weight of *A. tricolor* plants. The incorporation of organic materials, except EDTA in El-Gabal El-Asfar soil, significantly increased the fresh and dry weight of the tested plants compared to the control treatment. The magnitude of increase varied according to the added organic materials and soil types. Among all organic materials, EDTA was the most effective material in Helwan soil. Meanwhile, the vinasse sugarcane (VSC) was the most effective one in El-Gabal El-Asfar and El-Madabeg soils. El-Gabal El-Asfar soil showed the highest dry matter of the *A. tricolor* plants than other soils. The highest increase of dry matter was observed in Helwan soil treated with EDTA followed by VSC that increased by about 4.13 and 3.97 times compared to the control treatment, respectively. Adding VSC to El-Gabal El-Asfar and El-Madabeg soils resulted in increasing dry matter of *A. tricolor* plants by 63.56 and 64.90%, respectively, compared to the control treatment. On the other hand, HA was the least effective in Helwan soil (18.18%), while the EDTA was the least effective in El-Madabeg soil (15.32%) compared to the control treatment.

### 3.2. Lead (Pb) Uptake by A. tricolor Plants

In all studied soils, the investigated organic materials significantly (*p =* 0.05) increased the Pb concentration in *A. tricolor* plants compared to the control treatment (Table 3). The degree of the uptake varied depending upon the organic material and soil type. The organic material differed in its behaviour according to soil types. HA gave the highest increase in Pb concentration in plants grown in Helwan soil, followed by VSC. EDTA showed the highest value in the plants grown in El-Gabal El-Asfar soil, whereas VSC induced the highest amount in El-Madabeg soils.

Meanwhile, the highest Pb concentration was observed in El-Gabal El-Asfar soil, followed by Helwan soil and finally El-Madabeg soil. Application of HA and VSC caused an increase (5.83 and 5.90 times, respectively) in the Pb concentration in Helwan soil. In El-Gabal El-Asfar soil, the concentration of Pb increased by about 65 and 9% for EDTA and PLE treatments, respectively. At the same time, the application of VSC and HA in El-Madabeg soil raised the Pb concentration by about 110 and 66%, respectively.

### 3.3. Zinc (Zn) Uptake by A. tricolor Plants

The studied organic materials induced uptake of Zn in *A. tricolor* plants over the control (Table 4). The increase in Zn concentration in plants varied according to both the organic materials and the soil types. EDTA was superior to rest treatments regarding the concentration of Zn uptake in plants in all studied soils. While VSC and PLE materials recorded the second-highest Zn in El-Gabal El-Asfar and El-Madabeg soils, respectively. Application of EDTA and HA in Helwan soil caused an increase in Zn concentration in *A. tricolor* plants by 7.74 and 24%, respectively. In the case of plants in El-Gabal El-Asfar soil, the concentration of Pb increased by 42.83 and 11.54% for EDTA and VSC treatments, respectively. While the application of EDTA and PLE raised the plant Pb concentration by 56.60 and 21.40%, respectively, in El-Madabeg soil. El-Gabal El-Asfar soil showed high shoot Zn concentration than the other two soils.

### 3.4. Copper (Cu) Uptake by A. tricolor Plants

The data presented in Table 5 showed that the organic materials significantly increased the uptake of Cu over the control. The increase in Cu uptake varied according to both the organic materials and the soil types. EDTA and HA were the most effective organic materials for increasing Cu concentration in the shoot in Helwan soil. Cu concentration in the shoot of *A*. *tricolor* plant increased to 32.53 and 32.71% for Helwan soil due to EDTA and HA, respectively. In the same context, EDTA was the most effective treatment in both El-Gabal El-Asfar and El-Madabeg soils. It caused an increase in the shoot Cu by 26.82 and 37.21% for El-Gabal El-Asfar and El-Madabeg soils, respectively. VSC recorded the second highest order in El-Gabal El-Asfar and El-Madabeg soil since it increased Cu by 31.15 and 15.11% over the control, respectively. On the other hand, PLE was the least effective organic material in enhancing amounts of Cu that were taken up by *Amaranthus* plants grown in El-Gabal El-Asfar and El-Madabeg soils.

### 3.5. Effect of Organic Materials on Pb, Zn, and Mn Phytoextraction by A. tricolor Plants

The total amount of metal removed from contaminated soils at the end of the remediation process is the only way to evaluate the effectiveness of the remediation process. The remediation factor (RF) was calculated to assess the ability of *A. tricolor* plants in Pb, Zn, and Mn phytoextraction. This parameter shows the level of metal removal from soil by the plant. In the case of Pb, the highest value of RF was recorded when *A. tricolor* plants treated with VSC in Helwan and El-Madabeg soil (5.8 and 6.40%, respectively), while the value was 2.7 under PLE treatment in El-Gabal El-Asfar soil (Figure 3). However, the VSC gave the highest RF value of 1.68 and 3.26% for Zn in El-Gabal El-Asfar and El-Madabeg soil, respectively. Meanwhile, the highest value of RF (1.67%) was observed with EDTA in Helwan. Except for EDTA in El-Gabal El-Asfar soil, the lowest RF value of Pb and Zn was observed of *A*. *tricolor* plants grown on the control soils.

## 4. Discussion

The dry biomass of plants increased in all tested organic materials compared to the control. The properties of biochar have a great impact on HM immobilization [41]. The increase in fresh and dry weight by applying the organic material could be explained by the fact that the presence of soil organic matter improves physical, chemical, and biological characteristics and the fertility of studied soils [30,31]. It was further supported by applying organic materials that showed enhanced soil fertility by improving soil physical, chemical, and biological properties, and biomass production of plant and uptake of heavy metals through altering the physiological and morphological characteristics [32]. EDTA was the most effective material in Helwan soil, while the Vinasse sugarcane (VSC) was the most effective one in El-Gabal and El-Madabeg soils. These results agreed with those obtained by [33], who postulated that organic amendments promoted switchgrass growth. It has been reported that the application of vinasse increased the yield of wheat [34,35,36]. Also, [28] revealed that some organic compounds in the soil increased both the fresh and dry yield of tomato plants.

Moreover, the enhanced uptake and accumulation of HMs such as Pb, Zn, and Ni in sunflower have been reported recently when adding biosolids and cow manure simultaneously [16]. Increasing the *A. tricolor* plants’ dry matter in El-Gabal El-Asfar soil may be due to the high organic matter content than other soils (Table 1). Helwan soil has been characterized by high salt content, low organic matter (Table 1) that probably caused stress on the plants grown on this soil compared to the other soils. In contrast, the direct adverse action of EDTA as salts on the dry weight of *A. tricolor* plants in El-Gabal El-Asfar soil might be decreased plant growth. As indirect action, increasing the bioavailability of soil HMs could negatively influence plant performance. So, the reversible effect of EDTA on the plant growth in El-Gabal El-Asfar soil may be due to increasing the bioavailability of the HMs in this soil that contains a high amount of metals. EDTA treatments reduced the shoot growth of sunflower and saltbush plants [8,37]. They reported that EDTA might chelate HMs that enable the plant to absorb many of them, leading to reduced dry matter. Also, this result is agreed with those obtained by [38], who indicated that the application of EDTA at a level of 5.0 mmol/kg soil or higher significantly decreased the plant growth (dry weight).

The phytoremediation action of plants mainly involves phytoextraction or phytostabilization strategies. The efficiency of plants to absorb and accumulate a large quantity of metal in their tissue in a short period defined as phytoextraction. The metal of the hyperaccumulator is defined as a plant that can accumulate 100 mg kg^−1^ in its tissues [14]. The obtained data indicated that the investigated materials increased metals uptake by *A. tricolor* plants in all studied soils. The plants grown on El-Gabal El-Asfar soil showed the highest values of Pb uptake than the other soils. This result may be due to its high metal content and/or low soil pH value that led to increased mobility and uptake compared to the others. Increasing the amounts of Pb taken up by plants in the studied soils could be arranged in the descending order of El-Gabal El-Asfar soil > El-Madabeg soil > Helwan that is compatible with their Pb content. On polluted soils, a strong association was observed mainly between heavy metal content in the soil and its content of plants [31]. EDTA treatment caused the highest Pb amount in El-Gabal El-Asfar soil and the highest Zn content in all studied soils. This may be because it chelates the metal in this soil with low acidity and contains a high Pb and Zn content. Application of chemical chelators such as triethylene glycol diamine tetraacetic acid (EGTA) and sodium dodecyl sulfate (SDS) was reported as promising materials to use in HMs accumulation in ornamental plants [30,42,43,44]. EDTA was reported to be superior in solubilizing soil Pb and Zn for root uptake and its translocation into the above-ground biomass due to its strong chemical affinity for metals [31). It is considered one of the strongest synthetic chelating agents for metals with more ability than the naturally occurring organic ligands [28]. The higher level of EDTA caused an increased uptake of Cd by 63% due to the formation of complexes with Cd, which increases its availability and roots to shoot translocation [8]. Similar results were found by [43]. HA led to the highest increment of Pb in Helwan soil. These results are consistent with those obtained by [43,45], who found that HA enhanced plant growth and improved root development led to an increase in the availability of metals. Also, Refs. [46,47,48] reported that the HA was effective in the evapotranspiration, gas exchange, and leaf uptake of nutrients. Nonetheless, VSC caused a greater increment of Pb (110%) in El-Madabeg soil. This increase may be due to the formation of soluble organometallic complexes via vinasse application and its ability to increase metal availability in soil by re-distributing them from unavailable to available forms and low pH (Table 2). A similar result was obtained by [8], who indicated that the application of VSC at a rate of 30 mL kg^−1^ soil increased (42%) the availability of Cd compared to the untreated soil. The dissolved organic matter binds to metals and forms stable aqueous complexes with metals such as Pb, Cu, Zn, and Mn [49]. El-Gabal El-Asfar soil showed the highest shoot Zn concentration than the other two soils. These findings might be due to low pH values in both soils and VSC that encourage the plant’s mobility and uptake of metals.

In general, ornamental plants selected for phytoextraction should be fast-growing, deep-rooted, and easily propagated [14]. Plant depends on one or more of the many mechanics for its resistance and/or tolerance to metallic stress such as selective absorption metals, metal retention in roots, bind to metal-related compounds within the cell, retaining the metal in the cell wall, and regulating the metal concentration inside the cell through cellular mechanisms [50]. When the remediation factor (RF) value ranging between 1 and 10, the plant is considered an accumulator, and higher than ten plants is considered a hyperaccumulator [51]. The results showed that the ability of *A. tricolor* plants to accumulate Pb and Zn varied with soil type. *A. tricolor*, irrigated with Loom-dye effluent water, showed high accumulations of Pb, Cd, Cr, Fe, Mn, Zn, and Cu [52], inferring its efficacy of phytoextraction capacity. Wang et al. [53] reported that functional divergence between indigenous plant species *A. tricolor* and invasive alien species *Amaranthus retroflexus* L. and explored the strong competitive intensity in *A. retroflexus* over *A. tricolor* under HM stress, inferring the impact of successful invasion process and adaptation to ecological selection pressure.

The result indicated that the VSC gave the highest Pb (6.40%) and Zn (3.26%) removed from El-Madabeg soil. This result may be because VSC can improve soil structure and water holding capacity, promoting the growth of *A. tricolor* plants [8,30). High salt content, low pH (4.5), and the dissolved organics of VSC (Table 2) could be the reasons behind its high efficiency in mobilizing soil metals. Although, the EDTA showed the highest amount of absorbed Zn than other treatments, and the removal of Zn in El-Gabal El-Asfar soil decreases the plant biomass. Both the direct adverse action of EDTA as salt and its role in the increasing soil HMs bioavailability could negatively influence the plant performance. So, the reversible effect of EDTA on the plant growth in El-Gabal El-Asfar soil may decrease the dry matter, resulting in the reduction of RF for Zn. Conclusively, the detailed understanding of detoxification mechanisms employed by each OP for location specificity, distribution, and deposition of HMs in the cellular compartments such as cell walls, vacuoles, and metabolically inactive tissues have great concern as which plays a vital role in decreasing the free HMs concentrations [54].

## 5. Conclusions

The use of OPs is an eco-friendly technique that tends to clean up and accumulates HMs from contaminated soils. The efficacy of phytoremediation depends upon soil characteristics, available microbial types, their population size, and the phytoextraction efficiency of the plant species. Our results inferred that the application of organic materials increases fresh and dry weight while the degree of increment varied depends on the organic materials and soil types. El-Gabal El-Asfar soil showed the highest dry matter of the *A. tricolor* plants than other soils. The application of organic materials on the studied soils significantly increased Pb, Zn, and Cu uptake by *A. tricolor* plants compared to the control treatment. The removed metal amount is affected by the type of organic materials and soil type. In general, VSC was the most effective material for removing tested metals, especially in Gabal El-Asfar and El-Madabeg soil, while EDTA was the best effective in Helwan soil. The *A. tricolor* plants had a good ability to remove Pb than Zn and Cu, which was more evident in El-Madabeg soil treated with VSC. The difference in the amount of each metal removed from the different soils confirms our hypothesis due to the variation of soil properties. The identification of plant traits that contributed to enhancing phytoremediation in OPs is critical in uplifting phytoextraction capacity. In addition, the research gaps of exploring site-specific performances of OPs in phytoextraction, interaction with different chelators and beneficial soil micro-organisms, development of genetically modified OPs with enhanced detoxifying capacities, the improved function of antioxidants in detoxifying HMs while uptaking through root and foliage are of particular need to be filled. Moreover, the in-depth studies on HM-induced health risks to the human being, mainly when handling OPs are further investigated to minimize health hazards. According to our current study, using ornamental plants to remove mineral pollutants such as lead, zinc and copper from contaminated soil is more advantageous to minimize human health hazards because they avoid direct intoxication that could happen through edible plants. Thus, attention needs to be paid to bioavailability and uptake mechanisms of HMs and detoxification pathways to improve phytoextraction capacity in OPs in future studies. Also, extensive field experiments are necessary, especially in polluted soils near industrial areas, using these materials, particularly VSC, which has unique properties, that can be available in huge quantities. The in-depth experiments on mechanisms in cellular level HMs uptake and accumulation, signalling, and molecular events in OPs leading to phytoextraction are limited and need to be further explored to understand.

## Figures and Tables

**Figure 1 molecules-26-03360-f001:**
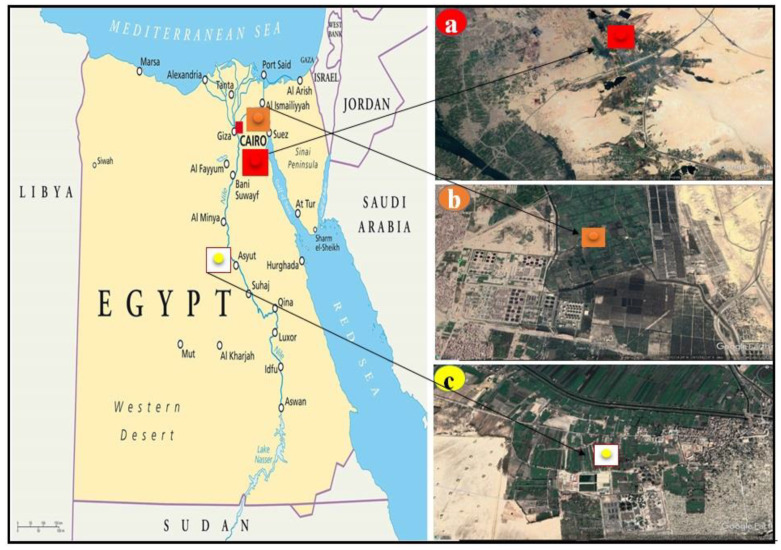
An aerial map of Egypt showing the location of the Helwan (**a**) El-Gabal El-Asfar (**b**) and El-Madabeg (**c**) soils scene acquired from Google Earth. 2020.

**Figure 2 molecules-26-03360-f002:**
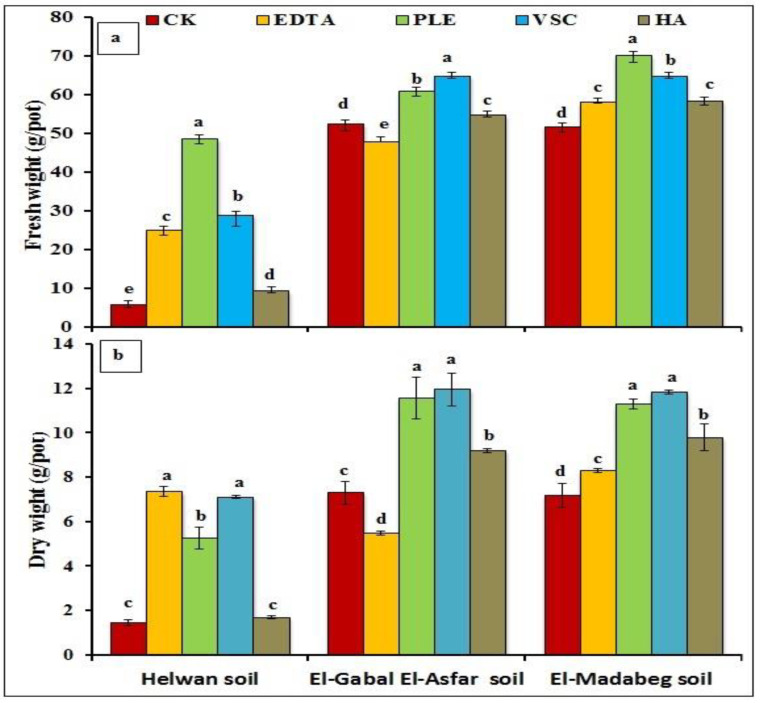
Fresh weight (**a**) and dry weight (**b**) of *A. tricolor* plants as affected by application of different organic materials.CK (control, No additions), EDTA (Ethylene diamine tetra acetic acid at 2 mM), PLE (poultry litter extract at 75 g/L), VSC (vinasse sugarcane 1:2 water) and HA (humic acid at 0.025%). Same letters were not significantly different at *p* < 0.05.

**Figure 3 molecules-26-03360-f003:**
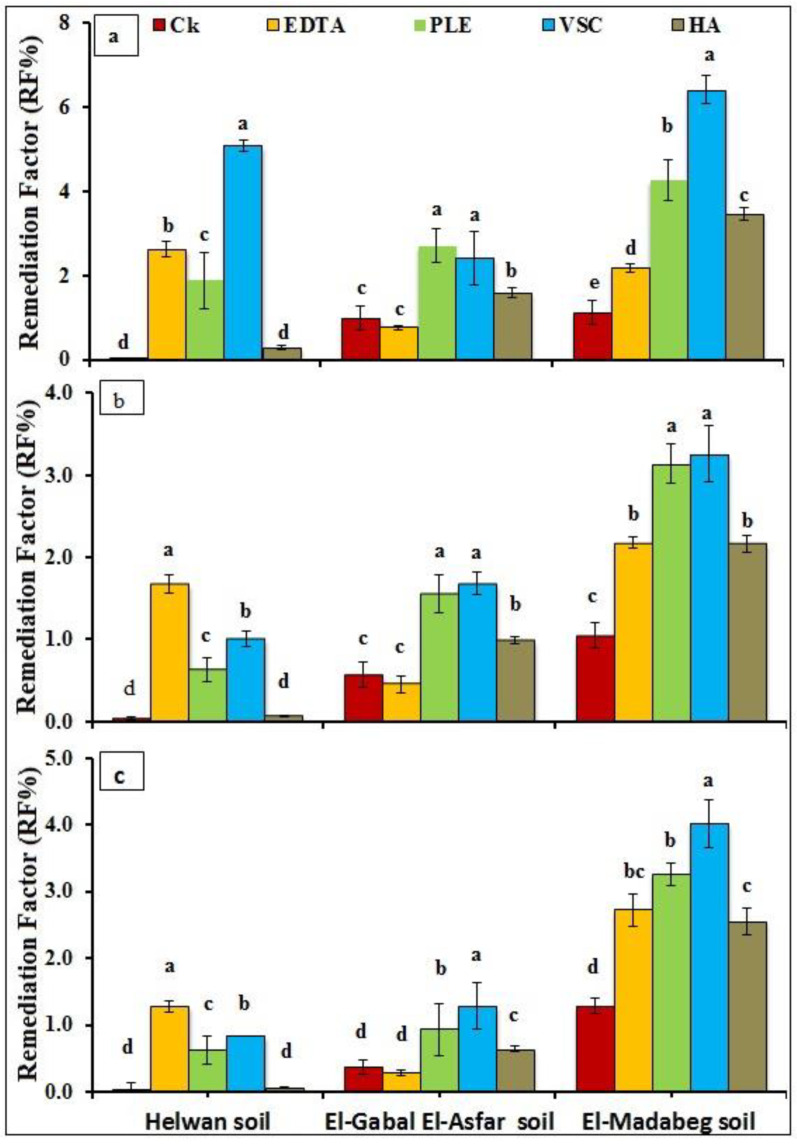
The remediation factor (RF%) of Pb (**a**), Zn (**b**) and Cu (**c**) by *A. tricolor* as affected by the application of different organic materials. CK (control, No additions), EDTA (Ethylene diamine tetraacetic acid at 2 mM), PLE (poultry litter extract at 75 g/L), VSC (vinasse 1:2 water) and HA (humic acid at 0.025%). Same letters were not significantly different at *p* < 0.05.

**Table 1 molecules-26-03360-t001:** Some chemical and physical properties of the studied soils.

Property	Soils		
	Helwan	El-Gabal El-Asfar	El-Madabeg
Clay (%)	24.74	12.00	4.91
Silt (%)	17.32	12.64	7.35
Sand (%)	57.94	75.36	87.74
Texture	Silty clay loam	Loamy sand	Sand
CaCO_3_ (g/kg)	53.7	25.0	68.00
pH (1:2.5)	8.11	6.71	7.59
Organic matter (%)	2.18	5.70	2.80
EC (1:1 dS/m)	5.18	1.86	1.7
US.EPA-extractable metals (mg/kg)			
Pb	45.60	247.20	86.10
Zn	202.90	585.30	139.10
Cu	36.10	191.20	38.30
Total metals (mg/kg)			
Pb	56.80	261.00	95.00
Zn	216.00	618.00	165.00
Cu	55.00	195.00	45.00
DTPA-extractable metals (mg/kg)			
Pb	2.70	45.51	6.16
Zn	11.91	5.63	4.10
Cu	3.21	3.75	6.07

**Table 2 molecules-26-03360-t002:** Some chemical properties of the examined organic materials.

Organic Material	Metals (mg kg^−1^)			EC (dS/m)	pH (1:2.5)	OM (%)
	Pb	Zn	Cu			
Poultry litter extract (PLE)	0.51	0.22	0.15	4.20	7.66	2.25
Vinasse sugarcane (VSC)	0.39	4.18	1.15	14.70	4.45	5.11
Humic acid (HA)	0.12	0.34	0.10	25.90	12.90	3.10

**Table 3 molecules-26-03360-t003:** Lead (Pb) concentrations (mg kg^−1^) in the shoots of *A. tricolor* grown in the contaminated soils amended with some organic materials.

Treatment	Helwan Soil	El-Gabal El-Asfar Soil	El-Madabeg Soil
CK	16.71 ± 0.27 ^d^	96.47 ± 0.46 ^d^	41.17 ± 1.08 ^e^
EDTA	55.43 ± 1.74 ^c^	132.69 ± 0.53 ^a^	60.53 ± 0.86 ^d^
PLE	77.18 ± 0.75 ^b^	105.33 ± 0.38 ^b^	63.41 ± 0.60 ^c^
VSC	114.23 ± 1.06 ^a^	88.04 ± 0.50 ^e^	86.77 ± 0.33 ^a^
HA	115.39 ± 0.57 ^a^	98.53 ± 0.57 ^c^	68.49 ± 0.11 ^b^

CK (control), EDTA (Ethylene diamine tetraacetic acid at 2 mM), PLE (poultry litter extract at 75 g/L), VSC (vinasse sugarcane 1:2 water) and HA (humic acid at 0.025%). Same letters were not significantly different at *p* < 0.05.

**Table 4 molecules-26-03360-t004:** Zinc (Zn) concentrations (mg kg^−1^) in the shoots of *A. tricolor* grown in the contaminated soils amended with some organic materials.

Treatment	Helwan Soil	El-Gabal El-Asfar Soil	El-Madabeg Soil
CK	84.93± 1.90 ^d^	129.97 ± 0.99 ^c^	67.01 ± 0.53 ^d^
EDTA	133.97± 1.13 ^a^	185.64 ± 1.18 ^a^	104.94 ± 0.93 ^a^
PLE	98.27± 1.62 ^c^	143.19 ± 1.10 ^b^	81.35 ± 3.06 ^b^
VSC	86.25± 0.77 ^d^	144.97 ± 1.04 ^b^	76.72 ± 1.71 ^c^
HA	105.53± 0.77 ^b^	144.55 ± 1.13 ^b^	74.65 ±0.59 ^c^

CK (control), EDTA (Ethylene diamine tetraacetic acid at 2 mM), PLE (poultry litter extract at 75 g/L), VSC (vinasse sugarcane 1:2 water) and HA (humic acid at 0.025%). The same letters were not significantly different at *p* < 0.05.

**Table 5 molecules-26-03360-t005:** Copper (Cu) concentrations (mg kg^−1^) in the shoots of *A. tricolor* grown in the contaminated soils amended with some organic materials.

Treatment	Helwan Soil	El-Gabal El-Asfar Soil	El-Madabeg Soil
CK	17.57 ± 0.57 ^c^	26.74 ± 0.44 ^c^	22.44 ± 0.57 ^d^
EDTA	26.04 ± 0.71 ^a^	36.54 ± 1.09 ^a^	35.74 ± 0.86 ^a^
PLE	24.68 ± 0.55 ^a^	27.09 ± 1.88 ^c^	23.02 ± 0.37 ^bc^
VSC	18.12 ± 0.25 ^b^	35.07 ± 1.01 ^a^	25.83 ± 1.51 ^b^
HA	26.11 ± 0.46 ^c^	29.35 ± 1.00 ^b^	23.76 ± 3.28 ^bc^

CK (control), EDTA (Ethylene diamine tetraacetic acid at 2 mM), PLE (poultry litter extract at 75 g/L), VSC (vinasse sugarcane 1:2 water) and HA (humic acid at 0.025%). Same letters were not significantly different at *p* < 0.05.

## Data Availability

The data is not publicly available, though the data may be made available on request from the corresponding author.

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
