# Peer review of "Ornamental Plant Efficiency for Heavy Metals Phytoextraction from Contaminated Soils Amended with Organic Materials"

_molecules, 2021, doi:10.3390/molecules26113360_

Round 1

Reviewer 1 Report

The manuscript ,,Ornamental plants efficiency for heavy metals phytoextraction from contaminated soils amended with organic materials,, addresses an important environmental issue. I propose a major revision before publication, but I am already checking this manuscript for the third time and, ultimately, it should be adopted after all the shortcomings have been remedied.

You can see my advice, comments and recommendations below.

The abstract should be slightly abbreviated and summarize the most important results.

Please be sure that your manuscript thoroughly establishes how this work is fundamentally novel. Specific comparisons should be made to previously published materials that have a similar purpose. Please present a strong case for how this work is a major advance. This needs to be done in the manuscript itself, not just in the response to review comments.

Please be sure that your abstract and your Conclusions section not only summarize the key findings of your work but also explain the specific ways in which this work fundamentally advances the field relative to prior literature.

The significance of this study should be more emphasize in the introduction.

Take a look at this paper, which may be helpful. https://www.sciencedirect.com/science/article/pii/S0013935121000748#!

Line 58: This very important paper would also deserve a place here. https://www.mdpi.com/2079-4991/11/4/861

Line 138 and 140: If possible, provide GPS coordinates of the location for better orientation.

Line 145, clay loam: Add this important reference. https://www.sciencedirect.com/science/article/pii/S0169131719301413

Line 150: The inscriptions on the figure are faintly visible. Please enlarge them.

Line 158: Indicate the molar concentration and purity of the chemicals used for the experiments.

Line 161: What form of clay is it more sodium or calcium clay? What percentage montmorillonite did this clay contain?  Montmorillonite have excellent adsorption capabilities.

Line 178: I am interested in measurement deviations. How many times have you taken measurements for each sample to make sure the results are correct.

Line 200: Check the correct notation of SI units in the entire manuscript.

Line 214: The notation of the equation with explanation should be improved.

Line 242: Figure 2 is low quality and blurred. Edit this figure to make it in good quality. I also recommend making graphs in color for better recognition of results.

Line 312: Figure 3 is also in poor resolution, please correct it and, if possible, enter the graphs in color.

Line 318, biochar section: It is necessary to add this very important paper to this place as a reference because it dealt with this issue it in great detail. This will improve the quality of this part. https://www.sciencedirect.com/science/article/pii/S0160412021001525

Line 414: Indicate the possible risks of such research. Add your recommendations for future research.

Line 465: Make sure the references are added correctly according to the journal's instructions.

Author Response

Reviewer #1:

Dear Sir

Thanks very much for your efforts and useful comments about our manuscript. We are very grateful and appreciate your good comments which help us to make the paper more quality and accurate. We responded to all the comments of the reviewers and heighted the changes in the manuscript, so anyone can follow the corrections. We have modified the manuscript accordingly, and the detailed corrections are listed below point by point and we hope this edition is more suitable and cover all the comments.

  1. The abstract should be slightly abbreviated and summarize the most important results.

Author response: We abbreviated it

  1. Line 138 and 140: If possible, provide GPS coordinates of the location for better orientation.

Author response: We add it in the text.

  1. Line 145, clay loam: Add this important reference. https://www.sciencedirect.com/science/article/pii/S0169131719301413

Author response: It is not possible to use research papers in this section because it is one of the specific properties of each soil (soil texture) and there is no similarity between one and the other and this classification was done according to the USDA taxonomy

  1. Line 150: The inscriptions on the figure are faintly visible. Please enlarge them.

Author response: We have made it clearer

  1. Line 158: Indicate the molar concentration and purity of the chemicals used for the experiments.

  1. Line 161: What form of clay is it more sodium or calcium clay? What percentage montmorillonite did this clay contain?  Montmorillonite have excellent adsorption capabilities.

Author response: Indeed, montmorillonite has high absorption capacities, but this study is not concerned with the quality of the clay, but is interested in describing the texture of the used soil and its content of the total amount of sand, silt and clay according to the USDA Soil Taxonomy.

  1. Line 178: I am interested in measurement deviations. How many times have you taken measurements for each sample to make sure the results are correct.

Author response: All the analyzes were made in three replication, and we have mentioned to this in more than one place in the throughout the manuscript as in a line 178, 185, 186, 190 and 209

  1. Line 200: Check the correct notation of SI units in the entire manuscript.

Author response: We checked it

  1. Line 214: The notation of the equation with explanation should be improved.

Author response: We improved it

  1. Line 242: Figure 2 is low quality and blurred. Edit this figure to make it in good quality. I also recommend making graphs in color for better recognition of results.

Author response: We change it to color graph

  1. Line 312: Figure 3 is also in poor resolution, please correct it and, if possible, enter the graphs in color.

Author response: We change it to color graph

  1. Line 318, biochar section: It is necessary to add this very important paper to this place as a reference because it dealt with this issue it in great detail. This will improve the quality of this part. https://www.sciencedirect.com/science/article/pii/S0160412021001525

Author response: We add it

  1. Line 414: Indicate the possible risks of such research. Add your recommendations for future research.

Author response: We add it

  1. Line 465: Make sure the references are added correctly according to the journal's instructions.

Author response: We checked all references

Reviewer 2 Report

The revised manuscript is improved and as such it can be considered for possible publication.

Author Response

Dear Sir

Thanks very much for your efforts and useful comments about our manuscript. We are very grateful and appreciate your good comments, which help us to make the paper more quality and accurate. 

Reviewer 3 Report

Dear authors

Please, find in attached file my suggestions, remarks and questions. Your study is interesting but some points remain unclear. It is therefore necessary to improve your manuscript before it is considered as suitable for publication in Molecules.

Author Response

Reviewer #3:

Dear Sir

Thanks very much for your efforts and useful comments about our manuscript. We are very grateful and appreciate your good comments which help us to make the paper more quality and accurate. We responded to all the comments of the reviewers and heighted the changes in the manuscript, so anyone can follow the corrections. We have modified the manuscript accordingly, and the detailed corrections are listed below point by point and we hope this edition is more suitable and cover all the comments.

  1. Please, find in attached file my suggestions, remarks and questions. Your study is interesting but some points remain unclear. It is therefore necessary to improve your manuscript before it is considered as suitable for publication in Molecules.

Author response: We have taken this into account, all comments - and we replayed all of them - included in the attached PDF. The main manuscript has also been modified according to that

  1.  

Reviewer 4 Report

The authors have assessed the potential of different organic amendments to enhance phytoremediation of metal polluted soil with an ornamental plant species. Overall, the study falls short of providing any novelty or relevant insights, presents many inaccuracies and mistakes, and denotes the authors' lack of familiarity with the topic. A few specific shortcomings:

Lack of novelty: Amaranthus tricolor has been extensively studied for phytoremediation purposes, including the use of different organic amendments. There are numerous papers covering the ability of this species for phytoremediation of metal polluted soil in the pertaining literature. Yet, the authors clearly state that "There is little information available on the use of A. tricolor plants in metal phytoremediation from the polluted sites".

Title: "Ornamental plants" - why plants if only one species has been studied?

Poor experimental design/lack of coherence: why would the authors employ "natural immobilized organic materials" (surely meant "immobilizing") in a phytoextraction study?

Poor experimental design/lack of coherence: Using EDTA is not only unacceptable, but also shows the authors lack of awareness on the topic. EDTA has been discarded as a valid amendment more than a decade ago, due to its hazardous effects and persistence in the environment. There are dozens of well-known, highly-cited papers demonstrating this. 

Poor experimental design/lack of coherence: Remediation Factor (RF) ?! Firstly, there are two essential factors in phytoremediation: bioconcentration and translocation factor. The RF calculated by the authors is a poorly calculated version of the bioconcentration factor that converts the concentrations to absolute amounts in shoot and soil. Secondly, the reference provided as support for the RF (Burt, R. Soil survey laboratory methods manual. Soil Survey Investigations Report No. 42, Version 4.0, Natural Resources Conservation Service, United States Department of Agriculture, 2004) does not mention any remediation factor!!! Third, hyperaccumulation cannot be determined via any of the mentioned factors. Once more, the authors show a complete lack of familiarity with the topic: "When the remediation factor (RF) value ranging between 1 and 10, plant is considered an accumulator, and higher than 10 plant is considered an hyperaccumulator".

Poor experimental design/lack of coherence: Contrary to what the authors clearly state, humic acid is not an "immobilizing" amendment. Quite often, humic and fulvic acids enhance the bioavailability of metals by promoting complexation.

Carelessness: In the abstract, the species is identified twice as A. bicolor

Results/discussion/conclusions: Due to critical mistakes in the experimental design (as explained above) the results, their discussion and conclusions are either flawed or superfluous. Again, the use of EDTA is not acceptable in this day and age, and the corresponding results are "old news". There are no relevant conclusions - what's more, half of the conclusions section is dedicated to further research.

Author Response

Reviewer #4:

Dear Sir

Thanks very much for your efforts and useful comments about our manuscript. We are very grateful and appreciate your good comments which help us to make the paper more quality and accurate. We responded to all the comments of the reviewers and heighted the changes in the manuscript, so anyone can follow the corrections. We have modified the manuscript accordingly, and the detailed corrections are listed below point by point and we hope this edition is more suitable and cover all the comments.

  1. Title: "Ornamental plants" - why plants if only one species has been studied?

              Author response: We have taken that into consideration.

  1. why would the organic employers "natural immobilized materials" (surely meant "immobilizing") in a phytoextraction study?

Author response: This study is part of a big study, and these materials have been subjected to study their effect on the movement of the studied minerals and soils, and it has been proven that they actually increase their mobility. Therefore, it was used in the phytoextraction study.

  1. Using EDTA is not only unacceptable, but also shows the authors lack of awareness on the topic. EDTA has been discarded as a valid amendment more than a decade ago, due to its hazardous effects and persistence in the environment. There are dozens of well-known, highly-cited papers demonstrating this.

            Author response: In this study, EDTA was used as an industrial material whose effect on heavy metal mobility was proven effective as a comparative material for the organic materials used

  1. Soil Survey Investigations Report No. 42, Version 4.0, Natural Resources Conservation Service, United States Department of Agriculture, 2004) does not mention any remediation factor !!! Third, hyperaccumulation cannot be determined via any of the mentioned factors.

Author response: Indeed, there was an error in the references that was corrected as the reference used in the remediation factor calculation is          ((Neugschwandtner, R.W.; Tlustoš, P.; Komárek, M.; Száková, J. Phytoextraction of Pb and Cd from a contaminated agricultural soil using different EDTA application regimes: laboratory versus field scale measures of efficiency. Geoderma, 2008, 144:446–454))

  1. Contrary to what the authors clearly state, humic acid is not an "immobilizing" amendment. Quite often, humic and fulvic acids enhance the bioavailability of metals by promoting complexation.

Author response: We have took that into consideration.

  1. In the abstract, the species is identified twice as A. bicolor

Author response: We correct it

Round 2

Reviewer 1 Report

The manuscript ,,Ornamental plants efficiency for heavy metals phytoextraction from contaminated soils amended with organic materials,, has been significantly improved. I recommend accepting in its current form.

Author Response

Dear Sir,

Thanks very much for your efforts and useful comments about our manuscript. We are very grateful and appreciate your good comments which help us to make the paper more quality and accurate. 

Reviewer 3 Report

Dear authors 

After your changes, I suggest two additions that you omitted in version 2. The first is related to the certified sample uses in the current study, the results obatined and the comparison with the certified values (Material and methods). The second is in relation to the metal exported.  "phytoextraction" in mentionned in the title of your manuscript. Consequently, if the remediation factor is important, data about the exported metal is crucial.

Author Response

Dear Sir.,

Thanks very much for your efforts and useful comments about our manuscript. We are very grateful and appreciate your good comments which help us to make the paper more quality and accurate.

  1. The first is related to the certified sample uses in the current study, the results obatined and the comparison with the certified values (Material and methods).

Author response: We consider that

  1. The second is in relation to the metal exported. "phytoextraction" in mentionned in the title of your manuscript. Consequently, if the remediation factor is important, data about the exported metal is crucial.

            Author response: We added the analysis of the used organic materials (Table 2), and this was considered when calculating the remediation factors

Reviewer 4 Report

Despite the authors effort, the manuscript remains unfit for publication. As detailed in the first revision, the study is fundamentally flawed at the experimental level. Moreover, the authors failed to adequately address the original remarks. In addition, different results strongly suggest that the statistical analysis is also incorrect (e.g. on Figure 3, certain significant differences, as identified by the lowercase letters, are very hard to believe). There are still many grammar/syntax mistakes.

Author Response

Dear Sir.

Thanks very much for your efforts and useful comments about our manuscript. We are very grateful and appreciate your good comments which help us to make the paper more quality and accurate.

  1. Different results strongly suggest that the statistical analysis is also incorrect (e.g. on Figure 3, certain significant differences, as identified by the lowercase letters, are very hard to believe).

Author response: We appreciate the accuracy of your comment, and we have modified the Figure

  1. There are still many grammar/syntax mistakes.

            Author response: We corrected the language and syntax of the whole manuscript.

This manuscript is a resubmission of an earlier submission. The following is a list of the peer review reports and author responses from that submission.

Round 1

Reviewer 1 Report

Sorry, I made a review of this paper before on january (molecules-1094254) and my comments were not taken in consideration. 

The submitted article is interesting and suitable for their publication, although I'm not sure if for this journal https://www.mdpi.com/journal/molecules/about In my opinion, it should be sent for environmental science or plant/soil science journal and not for this journal.

The scientific novelty of this paper is low. In my opinion, it should be assessed the available contents in studied soils after plant harvesting or even the relation between the plant extraction and the different soil extractants (e.g. https://doi.org/10.1016/j.jenvman.2019.01.058). Besides, several information is missing, discussion is too poor, etc.

I make again the same comments but updated to the new lines.

L31, 35. "tricolor"

L59. Please, be careful with formating because references numbers are not between brackets.

Introduction section. Before, I suggest several papers to improve this section. Please, improve the introduction (in the present version does not has quality!) Please see some papers: 
https://doi.org/10.1371/journal.pone.0095218 
https://doi.org/10.1016/j.ecoenv.2020.111836 
https://doi.org/10.1016/j.ecoenv.2020.111160 
https://doi.org/10.1007/s11356-020-08377-0
https://agris.fao.org/agris-search/search.do?recordID=AU2019115628
https://doi.org/10.1007/s11356-014-2947-z 
https://doi.org/10.1080/03601230802062273 
https://doi.org/10.3390/ijerph13030289 
https://doi.org/10.1016/j.iswcr.2015.07.001
https://doi.org/10.1007/s10661-015-4533-3 
https://doi.org/10.1016/j.chemosphere.2014.06.024

I suggest to improve these questions. Besides, it should be indicated that an explanation can be that A. tricolor is a plant specie with a concrete distribution (Asia) https://www.cabi.org/isc/datasheet/112199#:~:text=Distribution,-Top%20of%20page&text=Top%20of%20page-,A.,groups%20(Fiji%20etc.) and can explain this low number of papers, but other papers were also published with Amaranthus genera https://doi.org/10.1016/j.ecoenv.2012.01.015

Also, these papers can be useful for the discussion section.

Please, take care of these details.

L105-113. Please, give more details about the origin of the studied soils. Industrial soils? Ok, but what kind of industrial contamination?

L114. Put the data from clay, silt and sand as %. Texture classification... USDA? The same question for OM (% and not gkg).

According to your soil extraction, I think that it's a pseudototal content and not a total content (HF). Please, improve this question. Besides, please, put the first pseudototal and after available contents. Please, see the DTPA extraction that you have Pb, Zn and Zn and Cu is missing.

In addition, "US.EPA-extractable metals" is not referred to in the material and methods information. Is USEPA extraction? Please, give details. Besides, how were measured? ICP-OES, ICP-MS? Please, improve this section. A lot!

L128. Please, OM as %

L139. Why H2SO4 and HClO4 and not HNO3 + H2O2? Why not were differentiated between root ans shoots to see the translocation factor? See, for example https://doi.org/10.1016/j.jenvman.2019.01.058 Besides; I think that can be calculated the Phytoextraction potential (also from the reference 25), translocation factor or extraction efficiency (https://doi.org/10.1016/j.jenvman.2019.01.058). It should be interesting to assess the translocation factor soil-root and root-shoot.

L150. The number of replicates, etc.

- Figure 1 and 2. The quality figure should be improved. It's unreadable due to some issues with formatting.

- Table 3, 4 and 5 can be merged in a Table, and also a similar comment for these results section (can be merged in the same section).

- Besides the discussion is based on amendments, but not in other papers from the same genera. As I demonstrated before, there are more papers with similar approaches. The discussion section requires a big improvement.

Author Response

06 March 2021

Dear Sir,

Thanks very much for your efforts and useful comments about our manuscript. We are very grateful and appreciate your good comments that help us to make the paper more quality and accurate. We responded to all the comments of the reviewers and heighted the changes in the manuscript, so anyone can follow the corrections. We have modified the manuscript accordingly, and the detailed corrections are listed below point by point and we hope this edition is more suitable and cover all the comments.

Reviewer #1:

  1. L31, 35. "tricolor"

Author response: We correct it

  1. L59. Please, be careful with formating because references numbers are not between brackets.

Author response: we correct it

  1. L93-95. Please, give more details about the origin of the studied soils. Industrial soils? Ok, but what kind of industrial contamination?

Author response: we add these details

  1. Put the data from clay, silt and sand as %. Texture classification... USDA? The same question for OM (% and not gkg).

Author response: we change it

  1. Why H2SO4 and HClO4 and not HNO3 + H2O2? Why not were differentiated between root ans shoots to see the translocation factor? I think that can be calculated the Phytoextraction potential (also from the reference 25), translocation factor or extraction efficiency It should be interesting to assess the translocation factor soil-root and root-shoot.

Author response:; This study is part of a larger study, and in the case of estimating nitrogen, there is an overlap might occure in the N estimate, so we used this method.

And the interest in the transloction from the soil to the roots is not of our concern because we are interested in this study with the part that is above the soil surface because it is the edible part in most plants to be eaten

  1. In addition, US.EPA-extractable metals" is not referred to in the material and methods information. Is USEPA extraction? Please, give details. Besides, how were measured? ICP-OES, ICP-MS? Please, improve this section. A lot!

This method is referred to in the Methods and Materials section, line

Author response: we add it

  1. The number of replicates, etc

Author response: we mentioned it in line 154

  1. Figure 1 and 2. The quality figure should be improved. It's unreadable due to some issues with formatting.

Author response: we improved it

Reviewer 2 Report

The manuscript "Ornamental plants efficiency for heavy metals phytoextraction from contaminated soils amended with organic materials" is having interesting findings and solves an interesting topic. However a major revision is required before publication in MDPI -Molecules.

My advice, comments and recommendations are listed below:

The language of the manuscript should be more improved so that it is easy to read. You need to correct the grammar. Please go through the entire manuscript and shorten and correct some sentences.

I recommend clarifying and improve presentation of abstract so that it is clear to the reader what this is all about. Extend the abstract with your most significant results. Include your recommendations and future prospects.

Please be sure that your manuscript thoroughly establishes how this work is fundamentally novel. Specific comparisons should be made to previously published materials that have a similar purpose. Please present a strong case for how this work is a major advance. This needs to be done in the manuscript itself, not just in the response to review comments.

Please be sure that your abstract and your Conclusions section not only summarize the key findings of your work but also explain the specific ways in which this work fundamentally advances the field relative to prior literature.

The significance of this study should be more emphasize in the introduction.

See paper, which can help you: https://www.sciencedirect.com/science/article/pii/S0045653518323397

Line 58: Throughout the text, there is a problem with numbers and formatting references in square brackets. Fix it.

Line 60, section ,,The remedial methods,,: This statement confirms this very important paper and therefore it is necessary to add them to this place. https://www.sciencedirect.com/science/article/abs/pii/S0304389420316149

Line 110: Explain the abbreviation DTPA in parentheses in the text when you first used it.

Line 114: I am interested in measurement deviations. How many measurements did you take for each sample?

Line 117: What was the purity and molar concentration of the chemicals used for your experiments. Explain it in the manuscript.

Line 128: Here, too, I am interested in the measurement deviations and also how many times you performed the measurement for each sample.

Line 134: Indicate the country of origin of each device on which you performed the experiments.

Line 140: Use the correct designation of the equations with an explanation in the whole manuscript. 

Line 142, section of clay: This statement confirms this very important paper and therefore it is necessary to add them to this place. https://www.sciencedirect.com/science/article/abs/pii/S0169131719301413

Line 150: Slightly expand and explain this section about what it is about.

Line 167: Figure 1 is missing. Complete it and make it in high quality and color.

Line 183, 198 and 212: Check all the data in Table 3 and 4 and 5. What were the measurement deviations, how many times was it measured?

Line 226: The presentation and quality of figure 2 must be significantly improved. It is inaccessible in its current form. Distinguish the graphs in color.

Line 230: Discussion: I suggest adding more numbers to the Results and Discussion section. The current two figures are very few. You must add at least one or two.

Line 304, Conclusions: Extend the conclusion more with all your most important findings. Indicate the possible risks of such research. Add your recommendations for future research.

Make sure the references are added correctly according to the journal's instructions.

Author Response

06 March 2021

Dear Sir,

Thanks very much for your efforts and useful comments about our manuscript. We are very grateful and appreciate your good comments that help us to make the paper more quality and accurate. We responded to all the comments of the reviewers and heighted the changes in the manuscript, so anyone can follow the corrections. We have modified the manuscript accordingly, and the detailed corrections are listed below point by point and we hope this edition is more suitable and cover all the comments.

  • The language of the manuscript should be more improved so that it is easy to read. You need to correct the grammar. Please go through the entire manuscript and shorten and correct some sentences

Author response: We improved all manuscript language.

  • I recommend clarifying and improve presentation of abstract so that it is clear to the reader what this is all about. Extend the abstract with your most significant results. Include your recommendations and future prospects.

Author response: We improved it

  • Please be sure that your abstract and your Conclusions section not only summarize the key findings of your work but also explain the specific ways in which this work fundamentally advances the field relative to prior literature.

Author response: We improved it

  • Line 58: Throughout the text, there is a problem with numbers and formatting references in square brackets. Fix it

Author response: we correct it

  • Please be sure that your manuscript thoroughly establishes how this work is fundamentally novel. Specific comparisons should be made to previously published materials that have a similar purpose. Please present a strong case for how this work is a major advance. This needs to be done in the manuscript itself, not just in the response to review comments

Author response: we add it

  • Line 110: Explain the abbreviation DTPA in parentheses in the text when you first used it

Author response: we add it

  • Line 114: I am interested in measurement deviations. How many measurements did you take for each sample?

Author response: we add it

  • According to your soil extraction, I think that it's a pseudototal content and not a total content (HF). Please, improve this question. Besides, please, put the first pseudototal and after available contents. Please, see the DTPA extraction that you have Pb, Zn and Zn and Cu is missing.

Author response: we change it

  • Line 128: Here, too, I am interested in the measurement deviations and also how many times you performed the measurement for each sample.

Author response: we mentioned it in line 138

  • Line 140: Use the correct designation of the equations with an explanation in the whole manuscript. 

Author response: we mentioned it in line 164

  • Line 150: Slightly expand and explain this section about what it is about.

Author response: we explainmentioned it

  • Line 167: Figure 1 is missing. Complete it and make it in high quality and color.

Author response: we add it

  • Line 183, 198 and 212: Check all the data in Table 3 and 4 and 5. What were the measurement deviations, how many times was it measured?

Author response: we cheked it

  • Line 226: The presentation and quality of figure 2 must be significantly improved. It is inaccessible in its current form. Distinguish the graphs in color.

Author response: we improved it

  • Line 304, Conclusions: Extend the conclusion more with all your most important findings. Indicate the possible risks of such research. Add your recommendations for future research.

    Author response: we extend it
  • Make sure the references are added correctly according to the journal's instructions.

Author response: we checked it

Reviewer 3 Report

Specific comments

The study deals with an interesting topic, testing phytoremediation potential of an ornamental species Amaranthus tricolor growing in soils polluted by heavy metals (Pb, Zn and Cu) originated from industrial and municipal sources in several locations in Egypt. Using pot experiment, authors compared the effectiveness of organic (poultry litter extract, vinass sugarcane and humic acid) and synthetic (EDTA) material amendments in soils and their effects on accumulation of heavy metals in plant shoots. As such, it falls within the broad scope of Molecules journal and could be recommended for possible publication, however there are several deficiencies which should be considered prior to acceptance for possible publication in the journal. They are same of them as follows:

Introduction

This section is mostly based on well-known information on soil remediation technics, however there is lack of information on examined species. Instead, authors highlighted the importance of the examined species in phytoremediation of soils polluted with Pb, Zn and Cu by quoting reference on different species and different polluting element “Particularly Amaranthus tricolor reported to have specific interest amongst other ornamental plants as it has specific mechanisms to solubilize Cd adsorbed to soil particles while acquiring higher growth rate and large biomass when grown at heavy metal contaminated soils [21].”

Cay, S.; Uyanik, A.; Engin, M.S.; Kutbay., H.G. Effect of EDTA and tannic acid on the removal of cd, Ni, Pb and cu  from artificially contaminated soil by Althaea rosea Cavan. Int. J. Phyt 2015, 17,568–574

This section requires consideration.

 Materials and Methods

2.1. Characterization of soil sites

This part is mostly dedicated to brief description of sampling procedure, however more information on study sites is needed particularly in characteristics of examined soils i.e. soil types because authors discussed and concluded that “The removed metal amount is affected by organic materials and soil type.” Likewise, it is not clear what are the study sites “Surface soil layer (top soil) (0-30 Cm) was collected from three different locations in Egypt (Helwan and El-Gabal El-Asfar, Cairo Governorate and El-Madabeg soil, Assiut governorate).”, and where are they located and what are the particular sources of heavy metals pollution. Map of locations with marked sources of pollution is very welcome. The duration of experiment is also important.

  1. Results

 3.2. Lead (Pb) uptake by A. tricolor plants

This subsection started with one general statement “The factors affecting the amounts of metal that are absorbed by a plant are those controlling by the concentration and speciation of the metal in the soil solution; the movement of metal from the bulk soil to the root surface; the transport of the metal from the root surface into the root, and its translocation from the root to the shoot (Alloway, 1995).”-this statement should be the part of introductory section.

3.3. Effect of organic materials on Pb, Zn and Mn phytoextraction by A. tricolor plants

The meaning of the following statements is not clear: ” The total amount of the metal removed from contaminated soils at the end of the remediation process is the only way to evaluate the effectiveness of the remediation process. The remediation factor (RF) was calculated to assess the ability of saltbush plants in Cd phytoextraction. This parameter shows the level of metal removal from soil by the plant.”-this part should be considered and revised accordingly.

Discussion

Similarly to Results section in which obtained data were simply presented, results from this experiment were also linearly discussed, namely the linear relationship on Pb, Zn and Cu concentrations between three different soils/amendments were used. Lack of testing of mutual relationships between sources of pollutants/heavy metals and soil properties as well as characteristics of organic amendments is compensated with speculations based on literature sources. In addition, sometimes, it is difficult to distinguish results obtained in this study with literature sources which are listed in this section without clear interrelationship: “The results showed that the ability of A. tricolor plants to accumulate Pb and Zn varied with soil type. A. tricolor, irrigated with Loom-dye effluent water showed high accumulations of Pb, Cd, Cr, Fe, Mn, Zn, and Cu [46] infereing its efficacy of phytoextraxtion capacity Wang et al [47] reported that functional divergence between indigenous plant species Amaranthus tricolor and invasive alien species Amaranthus retroflexus L. and explored the strong competitive intensity in  Amaranthus retroflexus over A. tricolor under HM stress, inferring impact of successful invasion process and adaptation to ecological selection pressure.” etc. –need careful consideration.

Conclusions

This section is weak. Concluding remarks seem not related to the subject of the study. For instance, following statement: “It could be applied in catchment  areas where HMs contamination has become an unavoidable factor The use of ornamental plants to remove mineral  pollutants from contaminated soil is more safe compared to edible plants.” Such comparison has not been a subject of the experiment-need revision.

There are also parts which requires careful consideration such as following conclusion: “the OPs such as A. tricolor can be used as a hyperaccumulator in the uptake of HMs from the contaminated soils to reduce the metal load in the food chain.”Authors should carefully check the literature on hyperaccumulator plants particularly on concentrations of heavy metals they can accumulate in their tissues to be considered as hyperaccumulators and used in soils phytoremediation.

This section should also contain concluding remarks on scientific relevance and contribution of of the results obtained in this study to the existing knowledge of the subject.

Author Response

06 March 2021

Dear Sir,

Thanks very much for your efforts and useful comments about our manuscript. We are very grateful and appreciate your good comments that help us to make the paper more quality and accurate. We responded to all the comments of the reviewers and heighted the changes in the manuscript, so anyone can follow the corrections. We have modified the manuscript accordingly, and the detailed corrections are listed below point by point and we hope this edition is more suitable and cover all the comments.

Round 2

Reviewer 2 Report

The authors omitted some important recommended comments and corrections. They didn't even comment on them. For example:

The significance of this study should be more emphasize in the introduction.

See paper, which can help you: https://www.sciencedirect.com/science/article/pii/S0045653518323397

Line 60, section ,,The remedial methods,,: This statement confirms this very important paper and therefore it is necessary to add them to this place. https://www.sciencedirect.com/science/article/abs/pii/S0304389420316149

Line 142, section of clay: This statement confirms this very important paper and therefore it is necessary to add them to this place. https://www.sciencedirect.com/science/article/abs/pii/S0169131719301413

and other.

They replied to some comments that they had done so, but this is not visible in the revised manuscript. 

The authors should go through the whole review again and comment on all points.

Unless the answers and adjustments are complete, I cannot recommend accepting this manuscript.

Reviewer 3 Report

Specific comments

The revised manuscript is significantly improved, however there are still several deficiencies which should be considered prior to acceptance for possible publication in the journal. They are same of them as follows:

Introduction

In this section, there is an incorrect statement “Particularly Amaranthus tricolor reported to have specific interest amongst other ornamental plants as it has specific mechanisms to solubilize Cd adsorbed to soil particles while acquiring higher growth rate and large biomass when grown at heavy metal contaminated soils [21].”

Cay, S.; Uyanik, A.; Engin, M.S.; Kutbay., H.G. Effect of EDTA and tannic acid on the removal of Cd, Ni, Pb and cu  from artificially contaminated soil by Althaea rosea Cavan. Int. J. Phyt 2015, 17,568–574

This part requires correction.

  1. Materials and Methods

2.1. Characterization of soil sites

Information on study sites is still needed particularly in characteristics of examined soils i.e. soil types because authors discussed and concluded that “The removed metal amount is affected by organic materials and soil type.” Likewise, it is not clear what are the study sites “Surface soil layer (top soil) (0-30 Cm) was collected from three different locations in Egypt (Helwan and El-Gabal El-Asfar, Cairo Governorate and El-Madabeg soil, Assiut governorate).”, and where are they located and what are the particular sources of heavy metals pollution. Map of locations with marked sources of pollution is very welcome. Added statement “The soils at these locations are receiving a continuous supply of heavy metals as domestic, such as the soils of El-Gabal El-Asfar and El-Madabeg which for more than 50 years, and the Helwan soil, which are very close to the army factories that receive their industrial waste in addition to human waste.”, is not enough to explain the purpose of the study and offer an adequate information on sources of soil pollution.

Discussion

This section abundant in general and indirect statements on possible relationships between heavy metals accumulation in plants and properties of examined soils and organic amendments as well as phytoextraction potential of the examined plant species.

Conclusions

This section is improved however it is also abundant in many general recommendations for future research. It could be more useful to relate the phytoextraction potential of A. tricolor for phytoremediation of certain soils polluted with Cu, Zn and Pb in terms of soil type or source of the pollution.

This section should also contain concluding remarks on scientific relevance and contribution of of the results obtained in this study to the existing knowledge of the subject.